# Leveraging Generative AI and Large Language Models: A Comprehensive Roadmap for Healthcare Integration

**DOI:** 10.3390/healthcare11202776

**Published:** 2023-10-20

**Authors:** Ping Yu, Hua Xu, Xia Hu, Chao Deng

**Affiliations:** 1School of Computing and Information Technology, University of Wollongong, Wollongong, NSW 2522, Australia; 2Section of Biomedical Informatics and Data Science, Yale School of Medicine, 100 College Street, Fl 9, New Haven, CT 06510, USA; hua.xu@yale.edu; 3Department of Computer Science, Rice University, P.O. Box 1892, Houston, TX 77251-1892, USA; xia.hu@rice.edu; 4School of Medical, Indigenous and Health Sciences, University of Wollongong, Wollongong, NSW 2522, Australia; chao@uow.edu.au

**Keywords:** generative artificial intelligence, generative AI, large language models, LLM, ethics, healthcare, medicine

## Abstract

Generative artificial intelligence (AI) and large language models (LLMs), exemplified by ChatGPT, are promising for revolutionizing data and information management in healthcare and medicine. However, there is scant literature guiding their integration for non-AI professionals. This study conducts a scoping literature review to address the critical need for guidance on integrating generative AI and LLMs into healthcare and medical practices. It elucidates the distinct mechanisms underpinning these technologies, such as Reinforcement Learning from Human Feedback (RLFH), including few-shot learning and chain-of-thought reasoning, which differentiates them from traditional, rule-based AI systems. It requires an inclusive, collaborative co-design process that engages all pertinent stakeholders, including clinicians and consumers, to achieve these benefits. Although global research is examining both opportunities and challenges, including ethical and legal dimensions, LLMs offer promising advancements in healthcare by enhancing data management, information retrieval, and decision-making processes. Continued innovation in data acquisition, model fine-tuning, prompt strategy development, evaluation, and system implementation is imperative for realizing the full potential of these technologies. Organizations should proactively engage with these technologies to improve healthcare quality, safety, and efficiency, adhering to ethical and legal guidelines for responsible application.

## 1. Introduction

Since its release by the US company OpenAI in November 2022, a chatbot, ChatGPT, has stunned the world with its outstanding performance in conversation with humans [1]. Bill Gates acclaimed that the new generation of conversational agents ‘will change the way people work, learn, travel, get health care, and communicate with each other’, bringing in significant productivity improvement and reducing some of the world’s worst inequities, particularly for health [1]. The White House media release acclaimed that ‘From cancer prevention to mitigating climate change to so much in between, AI—if properly managed—can contribute enormously to the prosperity, equality, and security of all’ [2].

ChatGPT is a representative example of generative artificial intelligence (AI) technology. Generative AI refers to a subset of AI technologies that learn to predict the next word or sequence of words giving the preceding context. They can generate new content, such as text, images, music, speech, video, or code. Their huge success has attracted unprecedented speed of adoption, excitement, and controversy. Generative AI models use advanced deep learning and transfer learning algorithms and machine learning techniques to learn patterns and relationships from the existing data and generate new content similar in style, tone, or structure. Deep learning is a subset of machine learning that uses neural networks with multiple layers of processing nodes to analyze various factors of data for complex pattern recognition and prediction. Transfer learning is a machine learning technique that adapts a pre-trained model to a new but related task, leveraging knowledge from the initial task to improve new task performance.

Generative AI models are a subset of large language models (LLMs), e.g., generative pre-trained transformer (GPT). For example, GPT-3 is trained on 175 billion parameters, while GPT-4 is trained on one trillion parameters. An intermediary version, GPT-3.5, is specifically trained to predict the next word in a sequence using a large dataset of Internet text. It is the model that underpins the current version of ChatGPT [3]. After being pretrained on huge amounts of data to learn intricate patterns and relationships, these LLMs have developed capabilities to imitate human language processing [4]. Upon receiving a query or request in a prompt, ChatGPT can generate relevant and meaningful responses and answer questions drawing from its learned language patterns and representations [5]. These LLMs are often referred to as the ‘‘foundation model” or “base model” for generative AI, as they are the starting point for the development of more advanced and complex models.

Distinct from traditional AI systems, which are typically rule-based or rely on predefined datasets, generative AI models possess the unique ability to create new content that is original and not explicitly programmed. This can result in outputs that are similar in style, tone, or structure to the prompt instruction. Therefore, if designed thoughtfully and developed responsibly, generative AI has the potential to amplify human capabilities in various domains of information management. These may include support for decision-making, knowledge retrieval, question answering, language translation, and automatic report or computer code generation [4].

It is not surprising that a significant area for generative AI and LLM to revolutionize is healthcare and medicine, a human domain in which language is key for effective interactions for and between clinicians and patients [6]. It is also an information-rich field where every assessment, diagnosis, treatment, care plan, and outcome evaluation must be documented in specific terms or natural language in electronic health records (EHR). Once the LLM is exposed to the relevant EHR data set in a specific healthcare field, the model will learn the relationships between the terms and extend its model to represent the knowledge in this field. With the further advancement of generative AI technologies, including video and audio technologies, the dream is not far away for healthcare providers to audit instead of simply typing data into EHR. Clinicians may orally request computers to write prescriptions or order lab tests and ask the generative AI models integrated with EHR systems to automatically retrieve data, generate shift hand-over reports and discharge summaries, and support diagnostic and prescription decision-making. Therefore, generative AI can be ‘a powerful tool in the medical field’ [7].

Generative AI and LLMs have also sparked intense debates and discussions regarding their potential benefits, future perspectives, and critical limitations for healthcare and medicine. In Sallam’s seminal systematic review of 60 selected papers that assess the utility of ChatGPT in healthcare education, research, and practice, 85% (51/60) of papers cited benefits/applications, while an overwhelming 97% (58/60) raised concerns or possible risks associated with ChatGPT use [8]. These findings suggest that with proper handling of ethical concerns, transparency, and legal matters, these technologies could not only expedite research and innovation but also foster equity in healthcare.

To better harness the advancements in generative AI and LLMs, we conducted a comprehensive scoping review of the recent literature. The objective was to delineate a strategic roadmap for the effective integration and utilization of generative AI in healthcare and medicine. The guiding questions we used to find the literature were: What are generative AI and Large Language Models (LLMs)? What techniques are commonly used in the application of Generative AI and LLMs to healthcare and medicine? What are their current applications in healthcare and medicine? What are their benefits and unintended negative consequences? What are the ethical and regulatory considerations for these technologies? What are the future research and development trends for maximizing the benefits and mitigating the risks of integrating Generative AI and LLMs in healthcare and medicine?

## 2. Methods

The scoping literature review addressed questions a health or medical scholar without adequate machine learning background yet keen on the generative AI and LLM field might ask. To identify the pertinent literature, our primary search was structured into two steps using Boolean logic with the keywords listed in Table 1.

Step 1 aimed to grasp the scope of generative AI and LLM, initiating with Google Scholar because the significant articles that were pertinent to our inquiries, e.g., the development of Open AI’s GPT models and Google’s PaLM models, were published in arXiv, a free repository for academic pre-prints. One query often led to subsequent queries, guided by the referenced literature; therefore, we further assessed these references. Once well-informed about generative AI and LLM, we proceeded to Step 2, exploring the literature detailing their applications in healthcare or medicine in PubMed. The search period was from 1 March to 15 July 2023.

Article titles and abstracts were scanned to assess their relevance to our research questions. Noting the focus on GPT’s limitations and performance (see Table 2), we extended keywords in Step 3, addressing ethical and regulatory considerations for generative AI. Drawing from official websites, we compiled regulatory perspectives from the US and UK governments on generative AI. This iterative approach, utilizing the keywords in Table 1, resulted in two distinct concept clusters relevant to our enquiry from the 88 analyzed article titles and abstracts (see Figure 1).

Informed by the sharpened research questions and insights, we crafted the outline of our article, delineating fundamental research issues, concepts, and their causal interconnections. The iterative practices of evidence evaluation, question adjustment, and conceptual mapping persisted until we achieved satisfaction with the content. After this, we further polished and finalized the manuscript.

## 3. Results

Of the 63 papers included in this scoping review, 55 were academic papers providing a RIS file format. A concept analysis of these 55 articles revealed two concept clusters: one of which was centered on ChatGPT, model, patient, patient message, and research. The other emphasized response, physician, and question (see Figure 2).

We present our findings from seven aspects: technological approaches to generative AI applications, methods to train LLM, model evaluation, current applications of generative AI and LLM in healthcare and medicine, benefits, ethical and regulatory considerations, and future research and development directions.

### 3.1. Technological Approaches to the Application of Generative AI and LLMs

Generative AI and LLMs are powered by a suite of deep learning technologies. For example, ChatGPT is a series of deep learning models that utilize transformer architecture that resorts to self-attention mechanisms to process large human-generated text datasets (GPT-4 response, 23 August 2023). These AI technologies work in harmony to power ChatGPT, enabling it to handle a wide range of tasks, including natural language understanding, language generation, text completion, translation, summarization, and much more.

There are three key factors in choosing LLMs to solve practical problems: models, data, and downstream tasks [1], which also apply to solve healthcare and medicine problems.

#### 3.1.1. Models

Based on model training strategies, architectures, and use cases, LLMs are classified into two types [1]: (1) encoder–decoder or encoder-only language models and (2) decoder-only models. The encoder-only models represented by BERT family models have started to phase out after the debut of ChatGPT. Encoder–decoder models, e.g., Meta’s BART, remain promising as most of them are open-sourced, providing opportunities for the global software community to continuously explore and develop. Decoder-only models, represented by the GPT family models, Pathways Language Model (PaLM) introduced by Google [10], and LLaMA models from Meta, have and will continue to dominate the LLM space because they are the foundation models for generative AI technologies.

On the other hand, based on the training data set, LLMs are classified into foundation (or base) LLMs [11] and instruction fine-tuned LLMs [1]. The foundation LLMs, e.g., ChatGPT, are trained to predict the next most likely word that will follow based on the text training data; thus, the direction of output can be unpredictable. An instruction fine-tuned LLM is a base LLM to be fine-tuned, using various techniques, including Reinforcement Learning from Human Feedback (RLHF) [12]. As the instruction fine-tuned LLMs are better tuned to understand the context, input, and output in a specific application domain, they have improved their ability to align with purpose, overcoming the limitations in the base model, being safe and less biased and harmful. Therefore, instruction fine-tuned LLMs are the recommended LLMs to use in specific AI applications for healthcare and medicine [1]. This is supported by the finding of Singhal et al. that the instruction fine-tuned Flan-PaLM model surpassed its base PaLM model on multiple-choice medical question answering [6].

#### 3.1.2. Data

The impact of data on the models’ effectiveness starts from pre-training data and continues through to the training, test, and inference data [6]. The quality, quantity, and diversity of pre-training data significantly influence the performance of LLMs [1]. Therefore, pre-training base models on data from a specific healthcare or medical field to produce instruction fine-tuned models are the recommended development method for downstream machine learning tasks for these fields [13]. Of course, with abundant annotated data, both base LLM and instruction fine-tuned models can achieve satisfactory performance on a particular task and meet the important privacy constraint for healthcare and medical data [14].

#### 3.1.3. Task

LLMs can be applied to four types of tasks: natural language understanding (NLU), natural language generation, knowledge-intensive tasks, and reasoning [1]. Traditional natural language understanding tasks include text classification, concept extraction or named entity recognition (NER), relationship extraction, dependency parsing, and entailment prediction. Many of these tasks are intermediate steps in large AI systems, such as NER for knowledge graph construction. Using the decoder LLMs may directly complete inference tasks and remove these intermediate tasks.

Natural language generation includes two major types of tasks: (1) converting input texts into new symbol sequences, such as text summarization and machine translation, and (2) “open-ended” generation, which aims to generate new text or symbols in response to the input prompt, e.g., question answering, crafting emails, composing news articles, and writing computer codes [1]. This capability is useful for many tasks in healthcare and medicine.

Knowledge-intensive NLP tasks refer to tasks that require a substantial amount of background knowledge, whether it be specific knowledge in a particular domain, general real-world knowledge, or expertise gained over time [1]. These tasks not only require pattern recognition or syntax analysis but are also highly dependent on the memorization and proper utilization of knowledge about specific entities, events, and common sense of the real world. Healthcare and medicine tasks fit into this category. After exposure to a billion tokens, LLMs are excellent at knowledge-intensive tasks. However, in situations where LLMs have not learned the contextual knowledge or face tasks requiring this knowledge, LLMs would struggle and might “hallucinate” [11]. This problem can be solved by further exposing the LLM to the specific healthcare and medical knowledge base for retrieval augmentation to develop instruction fine-tuned models [15].

Reasoning tasks: In two seminal studies, Singhal et al. conducted a rigorous examination of the capabilities of LLMs by comparing their performance against both clinicians and laypeople in various cognitive tasks [6,16]. The LLM under investigation, Med-PaLM 2, demonstrated better alignment with a scientific consensus of 72.9% of the time compared to physician answers [16]. Med-PaLM 2 also achieved consistently high performance in tasks of comprehension (98.3%), knowledge recall (97.1%), and reasoning (97.4%). The metrics related to the potential for harm, as measured by the Agency for Healthcare Research and Quality (AHRC) common formats [17], suggested a significantly better performance of Med-PaLM2 (6.7% of answers) over clinicians (44.5% of answers) [16]. Med-PaLM exhibited reduced bias concerning medical demographics, registering such bias in only 0.8% of cases, as opposed to clinicians who exhibited bias in 1.4% of cases [6]. Med-PaLM was nearly on par with laypeople in crafting answers that directly addressed the user’s intent, scoring 94.4% versus the laypeople’s 95.9%. Remarkably, The answers from Med-PaLM 2 were preferred over those generated by physicians in eight of nine evaluation axes that pertained to clinical utility [16]. Thus, these experiments suggest that generative AI and LLMs can emulate human expertise in reasoning tasks.

The study also exposed areas where Med-PaLM underperformed clinicians [6], e.g., Med-PaLM lagged behind laypeople in generating helpful responses—achieving an 80.3% success rate compared to the laypeople’s 91.1%.

### 3.2. Methods to Train LLMs

#### 3.2.1. Fine-Tuning LLMs

LLMs can be fine-tuned by various strategies, e.g., modifying the number of parameters [18], size of the training data set, or the amount of computing used for training [1]. Fine-tuning LLMs will scale up the pretrained LLMs and significantly improve their performance in reasoning beyond the power-law rule to unlock unprecedented, fantastic emergent abilities [6,19]. Emergent abilities refer to specific competencies that do not exist in smaller models but become salient as the model scales. These include but are not limited to nuanced concept understanding, sophisticated word manipulation, advanced logical reasoning, and complex coding tasks [6]. For instance, when the PaLM model was scaled from 8 billion parameters to 540 billion parameters, it exhibited emergent abilities that essentially doubled its performance. The scaled Med-PaLM model achieved an accuracy of 67.2% in answering questions from the United States Medical Licensing Exam (USMLE) dataset.

Furthermore, the scaling of LLMs has led to advances that closely approximate human performance in both arithmetic reasoning and linguistic common-sense reasoning [1], competencies both important for healthcare and medicine. These enhanced capabilities allow LLMs to serve as innovative tools for medical education and help medical students gain novel clinical insights [20]. Moreover, the augmented abilities of LLMs in tasks involving recall, reading comprehension, and logical reasoning present opportunities for the automation of essential healthcare processes [6]. These may include, for example, clinical assessments, care planning, and the drafting of discharge summaries.

In addition, reinforcement learning from human feedback (RLHF) is a simple data- and parameter-efficient technique that can significantly improve the generalization capabilities and align LLMs to the safety-critical healthcare and medicine domain [21].

#### 3.2.2. Reinforcement Learning from Human Feedback (RLHF)

RLHF refers to methods that combine three interconnected model training processes: feedback collection, reward modeling, and policy optimization [22]. RLHF has been implemented as instruction prompts to train LLMs to achieve remarkable performance across many NLP tasks [6,16,18]. It not only improves model accuracy, factuality, consistency, and safety and mitigates harm and bias within medical question-answering tasks [6], but also bridges the gap between LLM-generated answers and human responses. Therefore, RLHF brings LLMs considerably closer to practical applications within real-world clinical settings.

#### 3.2.3. Prompt Engineering

Prompt engineering refines prompts for generative AI to generate text or images, often through an iterative refinement process. To date, five instruction prompts have been reported: zero-shot, few-shot, chain-of-thought, self-consistency, and ensemble refinement learning.

Zero-shot learning enables the training of LLMs for specific NLP tasks through single-prompt instructions, eliminating the need for annotated data [23]; e.g., people enter instructions into ‘prompt’ to seek answers from ChatGPT. This approach avoids the issue of catastrophic forgetting often encountered in fine-tuned neural networks, as it does not require model parameter updates [24]. Recent studies, such as those by Zhong et al., affirm the efficacy of LLM’s zero-shot learning in various traditional natural language understanding tasks [25].

Few-shot learning trains LLMs on specific NLP tasks by providing a limited set of example inputs, usually as input–output pairs termed “prompts” [16,26]. This learning technique facilitates quicker in-context learning compared to zero-shot learning, thereby producing more generalized, task-specific performance [6]. Umapathi et al. found that the level of performance improvement in hallucination control plateaued after three examples in a few-shot learning experiment [18]. They also found that the framing of prompts is crucial; concise and explicit prompts yield higher task execution accuracy compared to ambiguous or verbose ones.

Chain-of-thought prompting imitates the human multi-step reasoning process in problem-solving tasks. It enhances few-shot examples in the prompt with a series of intermediate reasoning steps articulated in concise sentences that lead to the final answer [6,16,27]. This method can effectively draw out the reasoning capabilities of LLMs [27] and shows substantial improvements in performance on math problem-solving tasks [28]. However, Singhal et al. did not find significant improvement in the performance of chain-of-thought prompting over the few-shot prompting strategy when applied to medical question-answering tasks [6].

Self-consistency prompting samples a diverse set of reasoning paths instead of only taking the greedy one [29]. Its logic is the common wisdom that a complex problem usually has multiple reasoning paths to reach the correct solution. It then selects the most consistent answer out of the sampled reasoning paths through unsupervised learning.

Ensemble refinement learning was introduced by Singhal et al. to improve the reasoning capabilities of the LLMs [16]. It takes a two-step approach: Step 1: give several (few-shot) chain-of-thought prompts and questions to randomly produce multiple explanations and answers (can vary temperature to increase sample size); Step 2: refine the model based on the original prompt, question, and the aggregated answers from Step 1 to produce a nuanced explanation and more accurate answer. Therefore, ensemble refinement prompting is a composition of chain-of-thought prompting and self-refining [30].

### 3.3. Model Evaluation

Three challenges impede the application of LLMs in modeling real-world tasks [1]: (1) noisy/unstructured real-world input data that are often messy, e.g., containing typos, colloquialisms, and mixed languages; (2) ill-defined practical tasks that are difficult to classify into predefined NLP task categories; and (3) ambiguous instructions that may contain multiple implicit intents. These ambiguities cause difficulty in predictive modeling without follow-up probing questions. Despite performing better than the fine-tuned models in addressing the above three challenges, the effectiveness of foundation models in handling real-world input data is yet to be evaluated [1,6]; therefore, Bommasani et al. calls for a holistic evaluation of LLMs [31].

Singhal et al. developed and piloted a seven-axes evaluation framework for the physician and lay user evaluation of LLM performance beyond accuracy on multiple-choice datasets [6]. The seven axes assess AI model answers for (1) agreement with the scientific and clinical consensus; (2) reading comprehension, retrieval, and reasoning capabilities; (3) incorrect or missing content; (4) possible extent and likelihood of harm; (5) bias for medical demographics; (6) laypeople assessment of helpfulness of answer; and (7) addressing the intent of the question. In the follow-up study, Singhal et al. added two additional human evaluations: (8) a pairwise ranking evaluation of model and physician answers to consumer medical questions along these nine clinically relevant axes and (9) a physician assessment of model responses on two newly introduced adversarial testing datasets designed to probe the limits of LLMs [16].

Kung et al. applied three model evaluation criteria: accuracy, concordance, and insight (ACI) to compare ChatGPT answers against those produced by two physicians [15]. They found that the accuracy of ChatGPT was strongly mediated by concordance and insight. High accuracy outputs were characterized by high concordance and high density of insight.

Liu et al. assessed the performance of their fine-tuned LLM CLAIR-Long, which is based on a LLM LLaMA-65B [32]. They employed four criteria, i.e., empathy, responsiveness, accuracy, and usefulness, to assess the responses from CLAIR-Long, ChatGPT, and four primary care physicians to patient queries received via an electronic health record portal. They found that both CLAIR-Long and ChatGPT performed well in terms of responsiveness, empathy, and accuracy despite a neutral result in usefulness. Based on this observation, they concluded that LLMs offer significant potential for improving communication between patients and primary care providers.

Chowdhury et al. examined the safety and appropriateness of zero-shot responses generated by ChatGPT to 131 unique questions asked by 120 postoperative cataract patients [33]. The evaluation tool was a simplified version of Singhal et al.’s (2022) human evaluation framework [6], with seven questions in three axes: intent and helpfulness, clinical harm, and clinical appropriateness. Two ophthalmologists independently used the tool to assess ChatGPT’s responses to the patient questions.

### 3.4. Current Applications of Generative AI and LLMs in Healthcare and Medicine

There is tremendous potential for LLMs to innovate information management, education, and communication in healthcare and medicine [7]. Li et al. proposed a taxonomy to classify ChatGPT’s utility in healthcare and medicine based on two criteria: (1) the nature of medical tasks that LLMs address and (2) the targeted end users [34]. According to the first criterion, seven types of ChatGPT applications were outlined: triage, translation, medical research, clinical workflow, medical education, consultation, and multimodal. Conversely, the second criterion delineates seven categories of end users: patients/relatives, healthcare professionals/clinical centers, payers, researchers, students/teachers/exam agencies, and lawyers/regulators.

A use case of LLMs to support the medical task of triage [34] is assisting healthcare professionals in condensing a patient’s hospital stay into succinct summaries based on their medical records, then generating discharge letters [35], benefiting from these models’ strong ability to summarize data from heterogeneous sources [36]. A useful application of LLM to improve clinical workflow is to significantly reduce the documentation burden that has long plagued doctors and nurses, a problem that persisted even after the transition from paper to electronic health records [37]. Importantly, LLM can improve interpretability [1], a vital goal in health data management. Therefore, they have the potential to lead to remarkable improvements in healthcare safety, quality, and efficiency.

Li et al. also stratified the 58 articles that they comprehensively surveyed into three distinct levels based on depth and sophistication [34]. Level 1 papers, including 41% of the articles (or 24 papers), offered foundational insights into either broad or specialized applications of ChatGPT in healthcare [34]. Level 2 papers, comprising 28% of the articles (or 16 papers), explored example use cases within specific medical domains and included a brief discussion about the accuracy of ChatGPT’s responses. Level 3 papers, making up the remaining 31% (or 18 papers), conducted qualitative or quantitative evaluations of ChatGPT’s responses to a comprehensive set of specialty or scenario-specific questions. This categorization indicates that the incorporation of generative AI into healthcare and medicine is currently situated within the incipient phases of the innovation diffusion process, specifically the “knowledge and persuasion” stage.

One paper classified under Level 2 explored ChatGPT’s utility as a support tool on breast tumor board decision making [38]. In this study, the researchers provided ChatGPT-3.5 with clinical information of ten consecutive patients presented in the breast tumor board of a medical center in Israel. Then, they asked the chatbot to conduct three tasks: summarization, recommendation, and explanation. Interestingly, the chatbot’s recommendations were congruent with the tumor board’s decisions in 70% of the cases. However, the chatbot did not recommend the inclusion of a radiologist in multidisciplinary consultations.

Level 3 papers predominantly focused on rigorous experiments to assess ChatGPT’s suitability for specific medical specialties or clinical scenarios [34]. Such assessments often employ Likert scale questions, varying from five-point to six-point to ten-point scales [39]. As of 30 March 2023, most of these Level 3 papers remained under review as pre-prints. Notably, most evaluations targeted ‘medical education’, which did not require ethical approval for experimentation [34].

Various studies have shown mixed results regarding ChatGPT’s efficiency. While ChatGPT demonstrated proficiency in fact-check questions, its performance on complex procedural questions was less consistent [40,41]. Sorin et al. endorsed ChatGPT as a reliable source for cancer-related or retina disease-related patient queries, but it had shortcomings in areas such as treatment recommendations [38]. Lahat et al. were able to use ChatGPT to identify the relevant research questions in gastroenterology but found that the answers lacked depth and novelty [39]. Duong emphasized ChatGPT’s aptitude for memorization and straightforward factual questions but noted limitations in tackling questions requiring critical thinking [42]. Consequently, Rao et al. advocated for the development of specialized AI tools to aid in complex problem solving in clinical workflow [40,41].

### 3.5. The Benefits of Generative AI and LLMs for Healthcare and Medicine

The application of generative AI and LLMs in healthcare and medicine remains predominantly within the academic research stage [43]. The following cases delineate preliminary efforts in the exploration of generative AI within these fields.

#### 3.5.1. Creating Synthetic Patient Health Records to Improve Downstream Clinical Text Mining

To date, many LLMs, e.g., ChatGPT, are only available through their APIs [14]. This raises privacy concerns for directly uploading patients’ data to LLM API for data mining. To tackle this challenge, Tang et al. propose a new training paradigm that first uses a small number of human-labeled examples for zero-shot learning via prompting on ChatGPT to generate a large volume of high-quality synthetic data with labels [14]. Using these synthetic data, they fine-tuned a local model for the downstream task of biological named entity recognition and relation extraction using three public data sets: BCBI Disease, BC5CDR Disease, and BC5CDR Chemical. Their training paradigm provides a useful application of LLMs to clinical text mining with privacy protection. It not only significantly reduces time and effort for data collection and labeling, but also simultaneously mitigates health data privacy concerns.

#### 3.5.2. Using Chatbot Underpinned by LLMs to Assist Health Communication

Ayers et al. evaluated the ability of ChatGPT to provide both quality and empathetic responses to patient questions [44]. They conducted a cross-sectional study to compare responses from ChatGPT and certified physicians to 195 patient questions posted on a public social media forum, Reddit. A team of licensed healthcare professionals carried out the evaluation, preferring chatbot responses above physician responses in 78.6% of the 585 evaluations. They rated chatbot responses with significantly higher quality and empathy. The study results suggest that AI chatbot assistants, with further review and approval from physicians, have the potential to aid in drafting responses to patient inquiries.

In contrast, Li et al. applied two strategies to surpass the known limitations of foundation LLMs, particularly their lack of specialized healthcare or medical knowledge, which could diminish the clinical utility of chatbots [43]. They started with training a generic conversation model on LLaMA-7B, using 52K synthetic data generated by instruction-following from Stanford University’s Alpaca project [45]. Subsequently, they fine-tuned this model using their collected dataset of 100,000 patient–physician conversations from an online medical consultation website and culminated in an online app ChatDoctor [43]. They implanted a “knowledge brain” within ChatDoctor, linking it with Wikipedia and/or offline medical-domain databases, to facilitate real-time information retrieval to answer medical questions. ChatDoctor outperformed ChatGPT in similarity metrics (BERTScores) of precision, recall, and F1 score. This enhancement significantly improved LLaMA-7B’s ability to understand patient inquiries and provide accurate advice, surpassing the foundation model ChatGPT.

#### 3.5.3. Potential to Address Routine Patient Queries following Routine Surgery

Chowdhury et al. tested LLM ChatGPT’s capability to safely address patient questions following cataract surgery. They sought answers from ChatGPT to 131 unique symptom-based questions posed by 120 patients and assessed the responses by two ophthalmologists [33]. Despite 21% of questions being unclear for answers, 59.9% of ChatGPT’s responses were rated ‘helpful’, and 36.3% ‘somewhat helpful’. A total of 92.7% of responses were rated as ‘low’ likelihood of harm, and 24.4% had the possibility of ‘moderate or mild harm’. Only 9.5% of answers were opposed to clinical or scientific consensus. Even without fine-tuning and minimal prompt engineering, LLMs such as ChatGPT have the potential to helpfully address real-world patient questions. Therefore, LLMs have the potential to helpfully address patient queries following routine surgery with further control of model safety.

#### 3.5.4. Improving Accuracy in Medical Image Analysis

A three-step approach employing a Generative Adversarial Network (GAN) was proposed to improve the resolution of medical images, a critical component in accurate medical diagnosis [46]. The proposed architecture was evaluated with four medical image modalities, utilizing four test samples drawn from four public data sets. The authors reported superior accuracy of the model’s output and image resolution. By achieving high-resolution medical images, this method has the potential to assist medical professionals in interpreting data more precisely, leading to an improvement in diagnostic accuracy and patient care.

#### 3.5.5. Potential to Provide Ongoing Clinical Decision Support throughout the Entire Clinical Workflow

Rao et al. tested LLM’s ability to provide ongoing clinical decision support [41]. They presented ChatGPT with a series of hypothetical patients, varied by age, gender, and Emergency Severity Indices (ESIs), and asked it to recommend diagnoses based on their initial clinical presentations. The test followed 36 published clinical vignettes from the Merck Sharpe & Dohme (MSD) Clinical Manual. The results were noteworthy: ChatGPT achieved an overall 71.7% accuracy across all 36 vignettes. Within that, it demonstrated a 60.3% accuracy rate in generating an initial differential diagnosis and reached its highest accuracy of 76.9% in making a final diagnosis. These findings provide evidence to endorse the integration of LLMs into clinical workflow, highlighting their potential to support clinical decision making.

#### 3.5.6. Fine-Tuning Local Large Language Models for Pathology Data Extraction and Classification

Bumgardner et al. introduced an innovative approach that utilized local LLMs to extract structured International Classification of Diseases (ICD) codes from complicated, unstructured clinical data, including clinical notes, pathology reports, and laboratory findings sourced directly from clinical workflows at the University of Kentucky [47]. Through fine-tuning, the researchers optimized a decoder model LLaMA, along with two encoder models, BERT and LongFormer. These models were then used to extract structured ICD codes, responding to specific generative instructions. The utilized dataset, consisting of 150,000 entries, included detailed pathology reports describing tissue specimen attributes, as well as final reports summarizing diagnoses, informed by microscopic tissue reviews, laboratory results, and clinical notes. The complexity was that individual cases might contain many tissue specimens.

Remarkably, the study found that the decoder model LLaMA 7b outperformed the encoder models BERT and LongFormer, even though the latter were trained with domain-specific knowledge [47]. This study contributes a valuable methodology for the effective integration of LLMs into real-world medical task execution and achieving this integration within the bounds of organizational policies and the established technical framework.

#### 3.5.7. Medical Education

Language is the key means of communication in healthcare and medicine [16]. It underpins interactions between people and care providers. A key application area of LLMs is medical communication and education for healthcare and medical student, staff, and consumers alike [6,15,16,46]. The Med-PaLM 2 model reached 86.5% accuracy in answering medical exam questions on the United States Medical Licensing Exam (USMLE) dataset [16]. Likewise, ChatGPT based on GPT-3.5 reached an accuracy of around 60% [15], and GPT-4 achieved 86.1% in the same USMLE exam [46]. Moreover, physicians preferred Med-PaLM 2 answers to those of the physicians on eight of nine axes about clinical utility [16].

The above results suggest that generative AI and LLMs can produce trustworthy and explainable outputs. They can serve as exemplary guides for human learners, particularly in drafting papers with high internal concordance, logical structure, and clear articulation of relationships between concepts [39]. They can also exemplify the deductive reasoning process [16]. Therefore, it appears both feasible and promising for LLMs to assist human learners in healthcare and medical education. Their integration into clinical decision-making processes could well be an achievable goal in the future.

### 3.6. Ethical and Regulatory Consideration for Generative AI and LLMs

The current mainstream view of the healthcare and medicine community towards LLM is a caution to balancing regulatory and technical challenges as the generative AI technology is still in the early experimental stage. For example, the well-known ChatGPT model is fine-tuned on Internet data instead of healthcare data. As the model output is impacted by the training dataset, the experts do not recommend direct utilization of ChatGPT without further specialization in healthcare or medicine [40]. It is well known that LLMs can generate outputs that are untruthful, toxic, hallucinated, or simply not helpful to the users [48]. Conversely, healthcare and medicine are safety-critical, “high stake” fields that simply cannot afford negative consequences [35]. Therefore, adequate attention needs to be paid to the following concerns with AI ethics: patient privacy and data security issues, bias in AI algorithms, and implications of AI-generated content in healthcare and medicine.

#### 3.6.1. Ethical Concerns

The large-scale use of ChatGPT has raised several social and ethical questions, e.g., the production of false, offensive, or irrelevant data that can cause harm or even threat to humanity, politics, warfare, and knowledge bases [7]. Training data patterns and algorithmic choices may reflect existing health inequalities [6]. Currently, the framework used for evaluating LLM application in healthcare is relatively subjective, limited by the current human knowledge and expert opinion, and lacks coverage of the full spectrum of the population. Another potential area of bias is the limited number and diversity of the human raters, i.e., clinicians and laypeople, who participated in the evaluation [6,16]. Harrer summarizes six ethical concerns for the use of generative AI technologies: accountability, fairness, data privacy and selection, transparency, explainability, and value and purpose alignment [7]. However, making significant progress on these problems can be a challenge since the detoxification method can have side effects [48]. For example, it can also cause LLMs to be more vulnerable to distribution shifts, thus reducing model performance on the language used by marginalized groups [49]. This can stem from the method exploiting spurious correlations in the initial toxic training data. Therefore, it is necessary to establish frameworks for the ethical release and use of generative AI applications [8] and standardize the list of actions required for generative AI and LLMs to address the ethical, technical, and cultural concerns of healthcare and medicine [7].

#### 3.6.2. Ensuring Patient Privacy and Data Security

The public, healthcare, and technology communities call for regulations and policies on data governance and privacy for AI technologies [7,50,51]. Currently, any data-entered prompts in ChatGPT are transferred to the server of the company OpenAI without any legal bounding, raising concerns about data privacy, which is not in compliance with personal data privacy legislations in many countries. Safeguarding AI systems and data is another critical concern for generative AI application in healthcare and medicine, which requires adequate data protection mechanisms to prevent unauthorized access as well as protection against adversarial cyber-attacks [52]. These concerns prohibit the direct use of commercial products, e.g., ChatGPT, in healthcare and medicine practice. They drive the research on the alignment of generative AI and LLMs with appropriate norms, values, and design goals [53].

#### 3.6.3. Addressing Biases in AI Algorithms

It is well-recognized that foundation LLMs can generate outputs that are untruthful, biased, toxic, hallucinated, or simply not helpful to users [48]. This is because the training objective is to predict the next token in a text instead of following the user’s instructions helpfully and safely [26]. For example, the OpenAI company recognizes that their InstructGPT model is biased towards the cultural values of English-speaking people as it is trained to follow instructions in English [48].

If the training data do not represent diverse populations or are not up-to-date, bias or even errors may be introduced, which may potentially exacerbate healthcare disparity and damage patient safety [54]. Therefore, Kung et al. suggest conducting robust AI failure mode analysis (e.g., language parsing error) to shed light into the etiology of inaccuracy and discordance [15]. Likewise, there is also the potential risk of the bias of the LLM evaluator towards the LLM-generated texts [36].

#### 3.6.4. Implications of AI Model “Hallucination” for Healthcare

An obvious limitation of LLMs is hallucination, which refers to the LLMs’ potential to generate information that, though plausible, may be unverified, incorrect, or false [48]. This impediment can cause serious consequences in healthcare applications [18], leading to inappropriate medical decisions that may compromise patient safety [38]. The fault may further have profound legal and liability ramifications.

Umapathi et al. conducted hallucination tests on the common LLMs, including Text-Davinci, GPT-3.5, Llama 2, MPT, and Falcon [18]. They assembled a new hallucination benchmark dataset, Med-HALT (Medical Domain Hallucination Test), by amalgamating multiple-choice questions and answers from medical examination tests across three countries—the USA, Spain, and India—and a Chinese region, Taiwan. They conducted two types of tests on Med-HALT, namely reasoning tests and memory-based hallucination tests, utilizing accuracy and pointwise scores as metrics. The latter accounts for the score as the sum of the positive scores for correct answers and imposes a negative penalty for incorrect ones.

Their findings indicate that Llama 2 and Falcon outperformed commercial variants such as GPT-3.5 and Text-Davinci in all hallucination tasks [18]. Specifically, Llama 2 70B outperformed other models with an accuracy of 42.21% and a score of 52.37 in the reasoning task of the false confidence test. Conversely, Falcon 40B excelled in the reasoning fake task, achieving an accuracy of 99.89% and a score of 18.56, illustrating its ability to distinguish between authentic and fabricated questions. In information retrieval tasks, Falcon models were better than OpenAI’s GPT models, with Falcon40B achieving the highest average accuracy across all tasks (43.46%). Nevertheless, none of these models attained an acceptable level of accuracy on the two tasks, stressing the persistent issue of hallucination within the current LLMs. The researchers also found that instruction tuning and RLHF had a negative effect on the model’s capacity to control hallucination.

The current approach of assessing the clinical knowledge of LLMs based on limited benchmarks is no longer sufficient to meet these complex demands [6]. Consequently, the continued investigation into understanding and mitigating hallucination within LLMs remains an essential endeavor in the integration of these technologies into healthcare applications.

### 3.7. Future Research and Development Directions

#### 3.7.1. Establishing Legislation, Policy, and Framework of Responsible AI

Governments across the world are at differing stages of maturation in their responses to the myriad opportunities and challenges brought forth by the cutting-edge generative AI technologies. Leading the charge, the US government has embarked on constructing a robust international framework to govern the creation and utilization of AI innovations, with a focus on principles that “keep Americans safe” [2].

A recent White House media release underlines three fundamental principles for responsible AI: safety, security, and trust. On 21 July 2023, President Biden convened representatives from seven leading AI companies—Amazon, Anthropic, Google, Inflection, Meta, Microsoft, and OpenAI—to solicit their voluntary commitments to responsible AI practices [2]. This marked a significant step towards forging a regulatory framework for responsible AI within the AI industry. These companies pledged to adhere to both internal and external security testing protocols before releasing their AI systems, sharing pertinent information within industry peers, governments, civil society, and academia on managing AI-related risks. Their commitments extend to investing in cybersecurity and insider threat, with a particular focus on protecting proprietary and unreleased model weights. The Biden–Harris Administration has indicated its intentions to formalize these principles with an executive order and to champion bipartisan legislation, affirming America’s leading role in responsible innovation. Moreover, the US government has undertaken consultations on these voluntary commitments with twenty countries, including Australia, Brazil, Canada, and the UK [2].

Parallel with the US, the UK government is actively engaged in a public consultation process concerning the nation’s AI regulatory framework, which is grounded on five principles: (1) safety, security, and robustness; (2) appropriate transparency and explainability; (3) fairness; (4) accountability and governance; and (5) contestability and redress [55]. These principles will be promulgated on a non-statutory basis and be executed by extant regulators, leveraging their domain expertise to tailor the application of these principles to the specific context in which AI will be employed. The UK government’s stance emphasizes a pragmatic, proportionate approach that fosters responsible innovation. It has adopted a deliberately agile and iterative approach, cognizant of the rapid pace at which these technologies are evolving.

There is a gap in research that evaluates LLMs’ utilization in healthcare and medicine, in particular, the issues related to homogenization and amplification of biases, as well as inherited security vulnerabilities from base models [31]. This underscores the multifaceted nature of responsible AI governance, which requires an integration of legal, ethical, and technical considerations.

#### 3.7.2. AI Alignment Research

Concerns about the ethical, legal, and risk-related issues associated with the implementation of generative AI technologies in practical settings have motivated AI alignment research. This line of research aims at aligning model outputs with the norms, values, and design objectives of their human developers and users [53,56]. The overarching goal is to build aligned models that are helpful, honest, and ethical [48]. However, the effort to align model outputs with the values of distinct user groups may instigate unintended negative consequences with potential societal ramifications for disparate groups [49]. The challenges are further exacerbated by the difficulties in utilizing automated metrics to comprehensively encapsulate intricate and subjective alignment goals [36]. An associated perplex is the trade-off of reducing performance on specific tasks, coined the “alignment tax” [48], and the imperative to consider the fluid and evolving nature of medical consensus [6].

Conventional methods of evaluating the clinical knowledge of LLMs through restricted benchmarks are inadequate for satisfying these multifaceted demands [6]. Meanwhile, alignment research that is exclusively focused on harm detection and reduction is fragmented and lacks the potential to foster significant advancement of technology [53]. Based on the philosophical examination of the fundamental elements of linguistic communication between AI chatbots and human respondents, Finlayson et al. identify and delineate optimal conversational norms for interactions between human and chatbots and recommend using these norms for aligning AI chatbots with human values across various sectors, including healthcare.

In the context of these reflections, alignment research will continue to be a major challenge for the generative AI community [57]. It calls for the creation of accountable and inclusive decision-making procedures, such as quickly discerning and communicating uncertainties in AI decision-making to users [5]. The realization of these objectives mandates interdisciplinary collaboration to customize alignment approaches across diverse domains and cultural landscapes [48]. A three-step human-centered generative AI development roadmap outlines this interdisciplinary collaborative AI development at three levels: aligning with human values, accommodating humans’ expression of intents, and augmenting humans’ abilities in a collaborative workflow [56]. To date, alignment research is a critical area of inquiry for the successful incorporation of generative AI technologies into the safety-critical healthcare and medicine fields.

#### 3.7.3. Reinforcement Learning with Human Feedback in Healthcare and Medicine

RLHF is a low-tax alignment method, whereby a well-crafted few-shot prompt may substantially enhance the entire model’s performance [48]. Therefore, a pivotal area for growth and innovation lies in the domain of RLHF, the formulation and development of tailored prompting strategies [56], and well-crafted reward functions [58] across diverse use cases. This approach will focus alignment research on the pressing need for AI integration in healthcare and medicine, given the distinct logical challenges and inherent risks of these fields.

It becomes imperative to devise innovative participatory and co-design methodologies in collaboration with domain experts [56]. Companies responsible for the creation of LLMs must strive to augment their models by maintaining continuous human engagement through the phases of system design, training, testing, and implementation [59]. This approach would ensure human oversight or co-piloting to optimize the technology’s efficacy, coupled with a forward-looking stance to remedy the existing limitations and further advance AI application within healthcare.

#### 3.7.4. Developing New Methods to Prevent Hallucination

The efficacy of the common LLMs, including Text-Davinci, GPT-3.5, Llama 2, MPT, and Falcon, fell short in a well-designed hallucination assessment [18]. Umapathi et al. found that these LLMs were too sensitive to prompt framing and decoding parameters. Even minor alterations in these parameters led to models that had previously produced accurate responses to start hallucinating with erroneous outputs. This phenomenon suggests a critical need for further research to improve the robustness of LLMs across diverse settings. Therefore, developing new methods to prevent hallucinations in model outputs is vital in ensuring the responsible use of AI within the safety-critical domains of healthcare and medicine.

#### 3.7.5. Developing New Evaluation Methods

The inherent breadth, complexity, and high stakes of healthcare and medicine necessitate the ongoing development of resilient evaluation methodologies and frameworks to assure LLM alignment within these specific domains [16]. This endeavor calls for interdisciplinary collaboration among AI researchers, clinicians, consumers, social scientists, ethicists, and policymakers. The application of various scientific and sociocultural viewpoints is essential to address the social and contextual ramifications, thereby ensuring equitable and responsible AI innovation [6].

A notable challenge associated with the use of similarity metric scores, such as BERTScores, to evaluate LLM outputs lies in a narrow focus on the resemblance to a golden standard (e.g., physician’s response), rather than assessing the accuracy or applicability of the generated content [32]. The process of human analysis, although critical, can be both labor-intensive and susceptible to inter-rater variability and bias [32,60]. There is also a pressing need to gauge the empathy encompassed within the responses generated by AI [44], and to develop rigorous rubrics facilitating human evaluation of LLM performance within real-world workflows and clinical scenarios [16].

There is a need for detailed pairwise assessments and adversarial evaluations to differentiate the performance of Med-PaLM 2 responses from those created by physicians [16]. Therefore, adversarial data can be systematically extended to encompass health equity and allow separate scrutiny of sensitive datasets. Additionally, the evaluation process should extend to multi-turn dialogue and frameworks for active information procurement.

Therefore, comprehensive evaluation methods and frameworks need to be developed to critically assess the performance, fairness, bias, and equity of LLM within the context of healthcare and medicine [6]. The evaluation process should incorporate diverse perspectives, patient values, and clinical use cases. Technical strategies must also be explored to efficiently identify and mitigate biases, contributing to the construction of a more nuanced and responsive framework for generative AI in the fields of healthcare and medicine.

#### 3.7.6. Application of the LLMs in Real-World Healthcare and Medical Settings

The 58 articles that Li et al. reviewed are all experimental reports [30]. Many reported studies are centered on the relatively straightforward tasks of multiple-choice medical question-answering [6,15,16,18], and none have reported on the actual deployment of ChatGPT in clinical settings. This absence reflects the evidence-based and rigorous nature of healthcare and medicine, where any technological implementation must be robustly designed, developed, and thoroughly tested before being employed by healthcare providers and consumers. Meanwhile, society’s heightened awareness of potential AI-associated risks, including bias, ethical considerations, and disinformation, cannot be overlooked [34].

With large volumes of free-text data available, LLM and generative AI demonstrate promising potential in healthcare applications [35]. The path forward necessitates further empirical studies to authenticate the efficacy of LLMs in practical healthcare and medical settings [16]. One key challenge in the clinical realm is the accurate collection and synthesis of patient information into assessments and care plans [6], making the creation of benchmark tasks reflecting genuine clinical duties and workflow an imperative avenue for upcoming research. Building on this, more randomized trials need to be conducted to appraise the utility of AI assistants in enhancing clinician–patient communication, alleviating clinician burnout, and improving patient outcomes [44].

## 4. Conclusions

This article examines the transformative potential of generative AI and LLMs in healthcare and medicine. It delves into the foundational mechanisms, diverse applications, learned insights, and the ethical and legal considerations associated with these technologies, highlighting the unique role of RLHF in model development. A limitation of this scoping review is its non-exhaustive nature, as it does not conduct a comprehensive, systematic appraisal of all the extant literature during the specific time period. The inclusion of numerous papers from arXiv that have not undergone rigorous peer review could potentially reduce the research’s rigor.

Unlike traditional rule-based AI, these contemporary technologies empower domain experts and necessitate collaborative co-design process involving both clinicians and consumers. Global efforts are centered on exploring numerous opportunities and challenges in areas such as ethics, transparency, legal implications, safety, and bias mitigation. The promise for improving healthcare quality, safety, and efficiency is significant. Healthcare organizations should actively engage with these technologies while upholding ethical standards.

## Figures and Tables

**Figure 1 healthcare-11-02776-f001:**
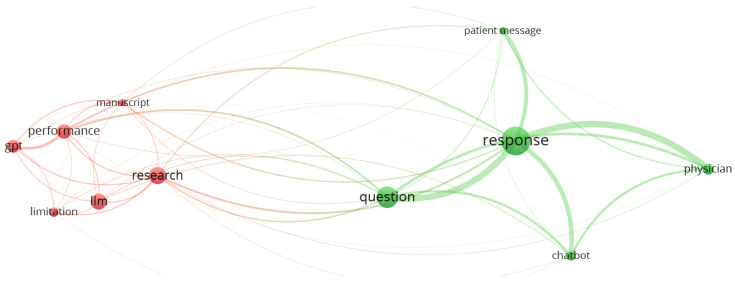
Visualization of two concept clusters featuring 13 pivotal concepts extracted from the titles and abstracts of the 88 sourced academic journal papers in a VOSviewer term map. VOSviewer is validated by Ref. [9].

**Figure 2 healthcare-11-02776-f002:**
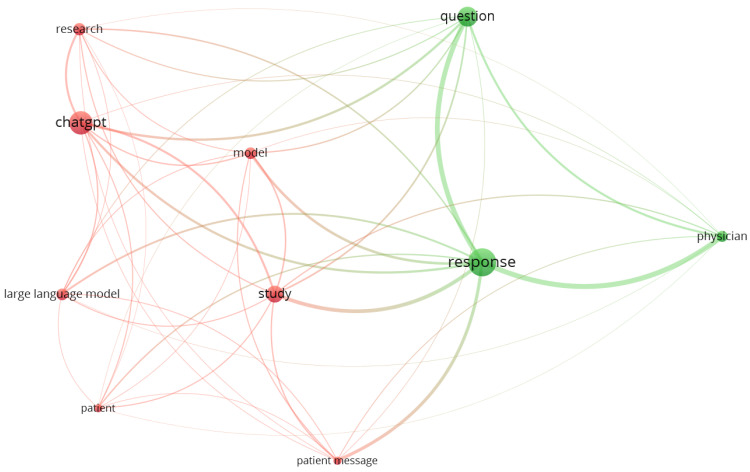
Visualization of two concept clusters featuring 10 pivotal concepts extracted from the titles and abstracts of the 55 reviewed academic journal papers in a VOSviewer term map.

**Table 1 healthcare-11-02776-t001:** The keywords used for the literature search.

Aim for Literature Search	Search Keywords
Step 1: Credible information about generative AI and LLM.	(“generative artificial intelligence” or “generative AI”) or/and (“large language model” or “LLM”)
Step 2: Application of generative AI and LLM in healthcare or medicine.	(“generative artificial intelligence” or “generative AI”) or/and (“large language model” or “LLM”) and ((“healthcare” or “health care”) or “medicine”)
Step 3: AI limitations, regulation, and ethics.	(“regulation” and “generative artificial intelligence” or “generative AI”) or/and (“large language model” or “LLM”) and ((“healthcare” or “health care”) or “medicine”) and (“limit*” or “align*” or “ethics”)

**Table 2 healthcare-11-02776-t002:** Comparison of key concepts included in the network of terms derived from the initially scanned 88 academic articles versus the 55 articles employed in this scoping review. Analysis was conducted on article titles and abstracts.

Terms from the Primary 88 Articles Scanned			Terms from the 55 Referenced Articles		
Term	^1^ Occurrences	^2^ Relevance	Term	Occurrences	Relevance
gpt	20	2.46	response	55	2.79
limitation	14	1.47	physician	17	2.73
performance	24	1.41	question	35	1.02
llm	27	1.13	patient message	11	1
response	58	0.85	chatgpt	44	0.86
patient message	11	0.81	research	19	0.59
physician	17	0.79	study	29	0.29
manuscript	10	0.79	large language model	18	0.26
chatbot	14	0.54	patient	11	0.24
research	30	0.48	model	18	0.22
question	40	0.28			

^1^ Occurrences: the number of times a specific term appears in the data set. ^2^ Relevance: normalized frequency of co-occurrence of the terms with the other terms in the data set.

## Data Availability

No new data were created or analyzed in this study. Data sharing is not applicable to this article.

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
