# Peer review of "Leveraging Generative AI and Large Language Models: A Comprehensive Roadmap for Healthcare Integration"

_healthcare, 2023, doi:10.3390/healthcare11202776_

Round 1

Reviewer 1 Report

Generative artificial intelligence (AI) and large language models (LLMs) are promising for revolutionising  healthcare and medicine.But while there is a great deal of research, what actually goes into the clinic has not yet been reported, and this contains a great deal of subject matter that can be discussed or debated. Being able to present the advances and concerns in this field in the form of a literature review will therefore be of interest to many professional researchers and readers interested in the direction of medical developments.

First of all, congratulations to the authors for completing such a comprehensive review!

There are a few suggestions that might make the paper better:
1. In the Methods section, which describes the methodology used for the literature review, it is suggested to be able to give a table or flowchart presentation in the Results section on the literature acquired, showing how much literature was included in this study.
2. "3.2 Method to train LLMs" also contains "3.2.3 Model evaluation", which may be misleading. The subheading of 3.2 could be modified appropriately.
3. "3.3 current......" The section "3.3 current " mentions the description of the literature at different levels, could it be better summarized in a table?
4. On page 11, line 550, it is not appropriate to refer to Taiwan as a country, which is a region of China.
5. different forms are used in the description of "LLaMA" and "LlaMA", and it is hoped that they can be standardized.

Reviewer 2 Report

This study conducts a scoping literature review to addresses the critical need for guidance on integrating generative AI and LLMs into healthcare and medical practices.Although, there are many suggestions in the current work.

Line 48 The authors can briefly mentioned which are the deep learning and transfer learning algorithms and machine learning techniques. 

The section on materials and methods should be a little more detailed. As example an Bibliometric analysis can be do with a map can be  generated from VOSviewer software.

A table with the key words used as he global semantic structure search.

A table summarizing the main findings of generative AI and its different applications in medicine, education etc. could be included in the results.

Include major limitations and future work.

The arXiv preprint references should be changed

The paper is mostly well organized, and the ideas are presented and described generally in an understandable manner. 

No comments
